# Design of Peptoid-peptide Macrocycles to Inhibit the β-catenin TCF Interaction in Prostate Cancer

Jeffrey A. Schneider[1], Timothy W. Craven[2,3], Amanda C. Kasper[2], Chi Yun[4], Michael Haugbro[2], Erica M. Briggs[1,5], Vladimir Svetlov[5,6], Evgeny Nudler[5,6], Holger Knaut[4], Richard Bonneau[3], Michael J. Garabedian[7,1], Kent Kirshenbaum[2] & Susan K. Logan[1,5]

New chemical inhibitors of protein–protein interactions are needed to propel advances in molecular pharmacology. Peptoids are peptidomimetic oligomers with the capability to inhibit protein-protein interactions by mimicking protein secondary structure motifs. Here we report the in silico design of a macrocycle primarily composed of peptoid subunits that targets the β-catenin:TCF interaction. The β-catenin:TCF interaction plays a critical role in the Wnt signaling pathway which is over-activated in multiple cancers, including prostate cancer. Using the Rosetta suite of protein design algorithms, we evaluate how different macrocycle structures can bind a pocket on β-catenin that associates with TCF. The in silico designed macrocycles are screened in vitro using luciferase reporters to identify promising compounds. The most active macrocycle inhibits both Wnt and AR-signaling in prostate cancer cell lines, and markedly diminishes their proliferation. In vivo potential is demonstrated through a zebrafish model, in which Wnt signaling is potently inhibited.

[1] Departments of Urology, New York University School of Medicine, New York, NY 10016, USA. [2] Department of Chemistry, New York University, New York, NY 10003, USA. [3] Department of Biology, Center for Genomics and Systems Biology, New York University, New York, NY 10003, USA. [4] Skirball Institute of Biomolecular Medicine, New York University School of Medicine, New York, NY 10016, USA. [5] Biochemistry and Molecular Pharmacology, New York University School of Medicine, New York, NY 10016, USA. [6] Howard Hughes Medical Institute, New York University School of Medicine, New York, NY 10016, USA. [7] Microbiology, New York University School of Medicine, New York, NY 10016, USA. These authors contributed equally: Jeffrey A. Schneider, Timothy W. Craven. Correspondence and requests for materials should be addressed to K.K. (email: kk54@nyu.edu) or to S.K.L. (email: Susan.Logan@nyulangone.org)

Protein–protein interactions (PPIs) provide the structural and functional basis for many critical biological processes. Extensive efforts have been made to identify small molecules or peptides capable of inhibiting particular PPIs that play a role in the pathogenesis of disease[1–3]. Despite these important advances, additional design strategies are urgently needed. Small molecules may lack sufficient surface area to abrogate protein–protein binding effectively, as these interfaces are typically broad and flat[2]. In addition, peptides often lack desirable pharmacological characteristics and are readily susceptible to degradation in vivo. These shortcomings can potentially be addressed by development of peptidomimetic oligomers that exhibit steric and chemical complementarity with protein surfaces[4–6]. Such molecules may populate an attractive middle ground between small molecules and biomacromolecular therapeutics. Crafting peptidomimetics suitable for targeting PPIs is a formidable and intriguing challenge. However, specific binding will require conformationally ordered oligomers that present diverse chemical groups in a predictable orientation.

Foldamers are sequence-specific oligomers that can mimic a range of biomolecular secondary structure motifs[7]. One particularly promising class of foldamers is peptoids, which are composed of sequences of *N*-substituted glycine monomer units. Although peptoids are oligo-amides, like native peptides, the side chains are presented from the backbone amide nitrogens instead of the α-carbons. This backbone modification confers peptoids with a broad resistance to proteolytic cleavage[8,9]. Peptoid synthesis efficiently employs the use of myriad primary amines as submonomer reagents, leading to rapid solid-phase generation of oligomers that exhibit extraordinary chemical diversity and tunable physicochemical properties[10,11]. These synthetic and structural features endow peptoids with the capability to satisfy the physicochemical requirements for a variety of biomedical applications.

Initially, attempts to establish conformational order for peptoids proved challenging, due to the lack of backbone hydrogen bonding, along with facile isomerization about the peptide backbone amide ω dihedral angle[12,13]. Strategies have now been identified to direct the organization of linear peptoid secondary structures, such as inclusion of bulky branched residue types[14–16]. Head-to-tail macro-cyclization can further rigidify peptoids in a β-hairpin conformation and can enhance biological activity[17–19]. Additional structural motifs can be attained in hybrid oligomer systems upon incorporation of α-amino acid residues[20]. This study evaluates the potential for peptoid-containing macrocycles to address outstanding challenges in molecular pharmacology, including targets that are currently considered undruggable.

The ability to employ computational design algorithms that can reliably identify particular foldamer sequences capable of binding a specific protein surface would provide a significant advance in targeting PPIs relevant to a wide array of disease states. Computational and experimental studies have demonstrated that there are discrete low energy conformations for cyclic peptides which are highly amenable to computational prediction[21–23]. Recent advances have elaborated protein design tools, such as the Rosetta molecular design suite to allow the modeling, docking, and sequence optimization of both peptide and peptidomimetic macrocyclic systems[19,23].

Among men, prostate cancer is the most prevalent form of cancer and the third most common cause of cancer-related death[24]. While patients with localized disease have a positive prognosis, those with metastatic disease have a 5 year survival rate of 30%. The main therapeutic target in prostate cancer is the androgen receptor (AR). Unfortunately, most patients treated with anti-androgens will develop resistance and continued disease progression, a state known as metastatic castration-resistant prostate cancer (mCRPC). Second generation anti-androgens, including Enzalutamide and Abiraterone, have been developed but extend life expectancy on average less than 6 months, as resistance can continue to develop[25,26]. There are multiple mechanisms of resistance, such as gene amplification or alternative splicing of the AR[27]. Additionally, the activation of auxiliary pathways can be oncogenic and/or upregulate AR signaling[28]. For instance, the AR gene is under transcriptional control of the Wnt signaling pathway[29,30]. The central Wnt regulator, β-catenin, can also interact with the AR and act as a co-activator[30,31].

Recent genome-wide sequencing studies found the Wnt signaling pathway to be mutated in upwards of 20% of tumors from patients with mCRPC, making it one of the most mutated pathways after AR, P53, and PTEN[32,33]. The most common Wnt mutations found were in APC or the phosphorylation domain of β-catenin, both of which cause the constitutive stabilization of β-catenin[34,35]. Following its stabilization, β-catenin translocates to the nucleus where it binds the T-cell factor (TCF) family of transcription factors. This causes a conformational change in TCF that activates transcription of a set of genes involved in cell proliferation and differentiation. β-catenin interacts with most of its binding partners including TCF, APC, Axin, E-cadherin, and the AR through its alpha-helical armadillo domain[31,36]. While the protein contacts partially overlap, there are distinct differences that can potentially enable specific inhibition of one PPI without affecting others. For example, a co-crystal structure of β-catenin and TCF shows there is a unique β-hairpin loop motif in TCF that binds to a cleft in the β-catenin armadillo domain[37].

Previously efforts have targeted the β-catenin:TCF interaction[38,39]. High-throughput screening has identified multiple classes of small-molecule β-catenin:TCF inhibitors[40–43]. These screens use either cell based luciferase assays or assays that monitor β-catenin:TCF interactions. Small-molecule virtual screening has also been used to identify inhibitors[44,45]. Some rational design techniques have been implemented that focused on modifying large peptide fragments associated with TCF's β-catenin binding domain in order to enhance their stability[46–48]. Nevertheless, there is still a need to establish a robust rational design strategy for discovery of β-catenin:TCF interaction inhibitors with promising pharmacological properties that can be advanced beyond pre-clinical testing.

In this study we apply computational protein design protocols to design macrocyclic peptoid–peptide hybrids to target the N-terminal TCF β-hairpin binding pocket of β-catenin and demonstrate its potential for use as a therapeutic in prostate cancer models. Using the Rosetta suite of computational tools, we first generate a small library of peptoid–peptide macrocycles designed in silico that are predicted to bind β-catenin. We use cell culture based luciferase assays to test the oligomers and select a promising compound. We show that the compound potently inhibits binding between β-catenin and TCF proteins. It also inhibits the proliferation of prostate cancer cells with nano-molar IC$_{50}$ values in both 2D and 3D cell culture models. In addition, we demonstrate that the peptoid macrocycle inhibits Wnt signaling in vivo through a zebrafish model.

## Results

**Design of oligomer macrocycles to bind β-catenin.** We identified a cleft on the surface of β-catenin created by armadillo repeats 8, 9, and 10 (Fig. 1) which was suitable for targeting with a cyclic oligomer. This region includes the binding site between β-catenin and the N-terminal sequence of *Xenopus laevis* TCF3, which forms a β-hairpin structure at the site of interaction in the X-ray structure (PDBid: 1G3J)[37]. The β-catenin binding domain

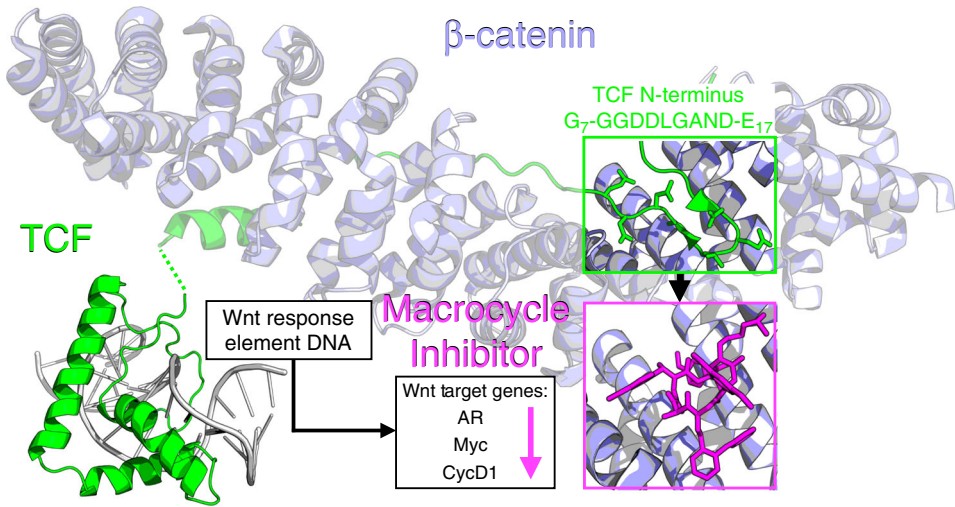

**Fig. 1** Proposed mechanism for targeting β-Catenin. The designed macrocycles inhibit the interaction of the N-terminal region of TCF and β-Catenin resulting in a downregulation of AR (Androgen Receptor), Myc, and CycD1 (Cyclin-D1)[37,72] Crystal structures depicted are PDB 1G3J and 2LEF

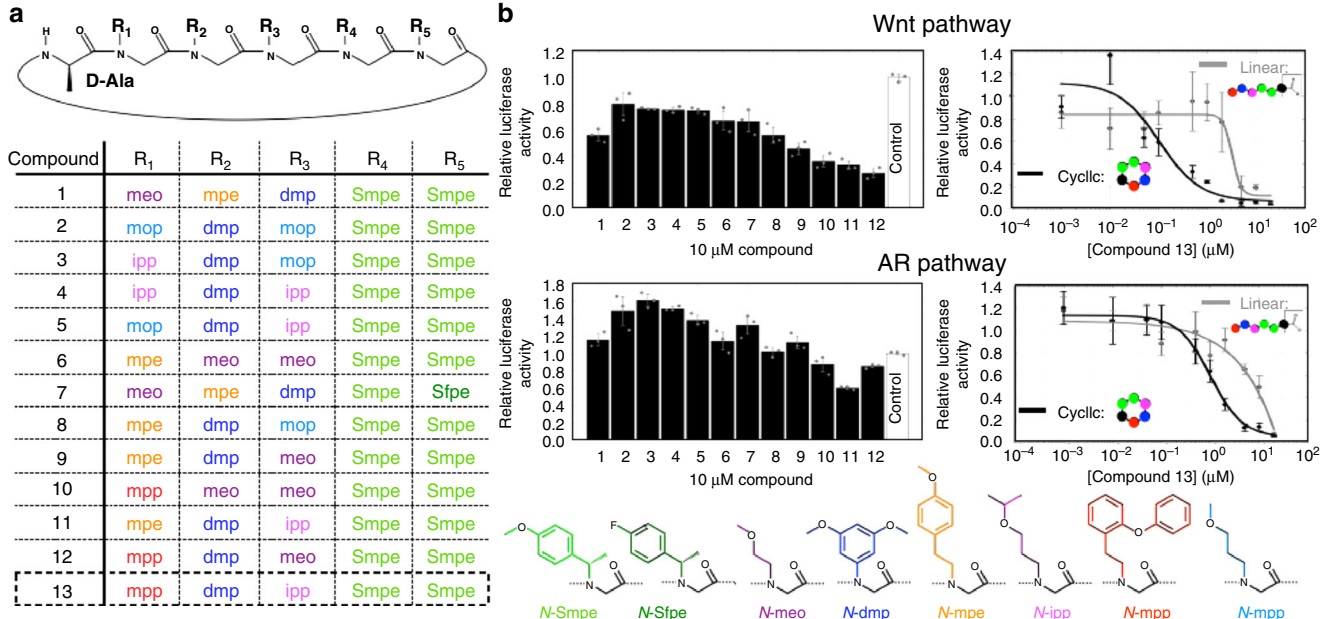

**Fig. 2** Design of oligomer macrocylces to bind β-catenin. **a** Peptoid–peptide hybrid macrocycle backbone scaffold. Middle table details compound number list and chemical compositions. Bottom section details the chemical compositions of peptoid side chains. **b** Luciferase reporter assay of compounds relative to a "vehicle" control. Data are presented as mean and SD ($n = 3$). Notes: "*N*-meo": *N*-(methoxy-ethyl)-glycine, "*N*-mpe": *N*-(4-methoxy-phenyl-ethyl)-glycine, "*N*-dmp": *N*-(3,5-dimethoxy-phenyl)-glycine, "*N*-Smpe": *N*-((S)-4-methoxy-phenyl-ethyl)-glycine, "*N*-Sfpe": *N*-((S)-4-fluoro-phenyl-ethyl)-glycine, "*N*-mpp": *N*-(2-phenoxy-phenyl-ethyl)-glycine, "*N*-ipp": *N*-(isopropoxy-propyl)-glycine, "*N*-mop": *N*-(methoxy-propyl)-glycine

of xTCF3 aligns very closely with the sequence of human TCF4 (Supplementary Fig. 1). This cleft is also adjacent to the $G_{13}ANDE_{17}$ region of the hTCF4 β-catenin binding sequence which had been mimicked in previous attempts to inhibit the β-catenin:TCF4 interaction[48,49].

We applied rational design techniques to target the xTCF3 β-hairpin binding pocket of β-catenin. In order to identify suitable oligomer scaffolds, we evaluated high-resolution structures experimentally determined for a set of peptoid and peptoid-α-amino acid hybrid macrocyclic structures[17,50]. One cyclic hexamer peptoid-α-amino acid hybrid (18-atom macrocycle Compound 1, Supplementary Fig. 2, Supplementary Table 1) was observed to provide a favorable steric complement to the targeted β-catenin cleft after modeling with the Rosetta molecular

design suite[51]. The in silico $\Delta G_{binding}$ values were evaluated after several iterations of guided (The PyMOL Molecular Graphics System, Schrödinger, LLC) and automated docking protocols utilizing Rosetta's InterfaceAnalyzer[52,53].

Inspection of the X-ray structure of **1** suggested that the D-Alanine and "*N*-Smpe" residues played a critical role in directing the overall fold of the oligomer. We hypothesize these bulky α-chiral peptoid side chains and a D-amino acid confer a unique structure to the oligomer backbone (Fig. 2a, positions colored green and black, respectively). In contrast, positions $R_1$, $R_2$, and $R_3$ appeared relatively amenable for substitution to optimize complementarity to the protein surface. After thorough analysis of **1** docked into the β-catenin X-ray structure (Supplementary Fig. 3), we selected a set of side chain types at each of the

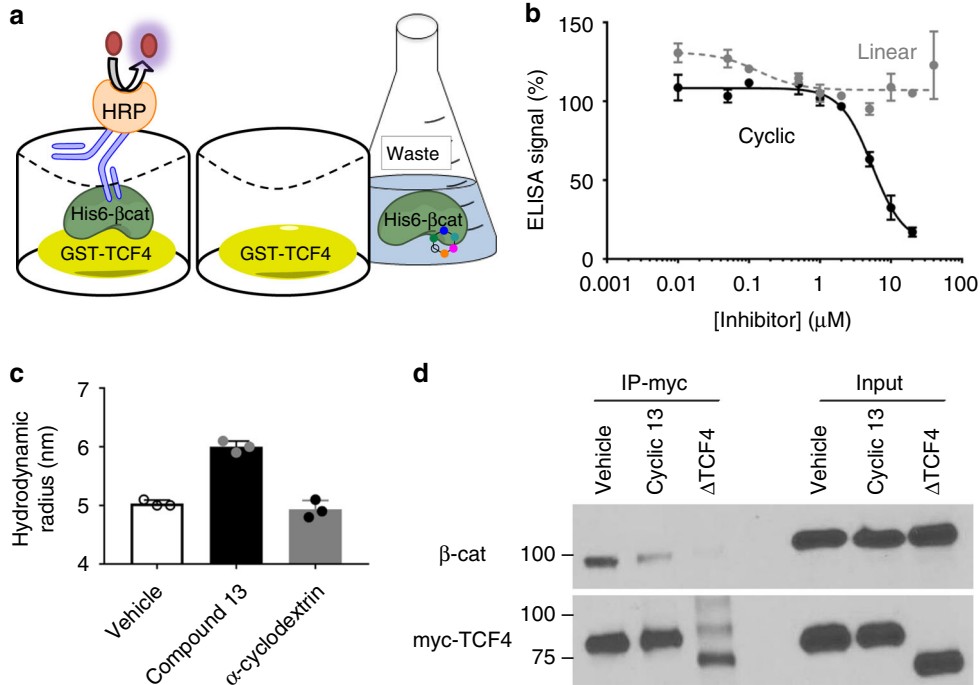

**Fig. 3** Macrocycle **13** inhibits the β-catenin:TCF4 interaction. **a** Schematic for ELISA based detection of recombinant His6-β-catenin binding to a GST-TCF4 N-terminal Catenin Binding Domain fusion protein. **b** Dose response curves for the cyclic oligomer (black solid line) and linear oligomer (grey dashed line) in the ELISA assay. **c** Dynamic light scattering measurements of the hydrodynamic radius of β-catenin in the presence of vehicle, compound **13**, or α-cyclodextrin. **d** Western blot depicting a co-immunoprecipitation (IP) for myc-TCF4 and β-catenin treated with compound. ΔTCF4: A myc-TCF4 construct that lacks the β-catenin binding domain. Data are presented as mean and SD ($n = 3$) of three independent experiments (**c**) or representative of three independent experiments (**b**, **d**)

designable positions based on a combination of visual inspection, modeling, aqueous solubility, and synthetic feasibility, as well as properties considered favorable for cell-permeability. A table of the shape complementarity and in silico $\Delta G_{binding}$ for each of the designs is provided in Supplementary Table 2.

**Compound testing and identification of a promising compound**. Using a TOP-Flash Wnt luciferase reporter, we tested the 12 oligomers identified from in silico modeling for inhibitory activity (Fig. 2b). At 10 μM concentration, all compounds had some level of Wnt inhibitory activity, with varying degrees of efficacy. We also employed a 3×-Androgen Response Element (ARE) luciferase reporter to measure the effect on AR activity. Only a few of the compounds were effective at inhibiting AR signaling, the best of which were also the most effective at inhibiting Wnt signaling. Based on the inhibitory data from other compounds containing similar side chain types, we reasoned that if we generated a macrocycle with the "N-ipp" side chain in position $R_3$ and the "N-mpp" side chain in position $R_1$ (Fig. 2a), we could create a more potent compound. An optimal sequence composed of these monomer types resulted in compound **13**. This compound was in fact more effective at inhibiting both the Wnt and AR luciferase reporters. A comparison of the luciferase activity inhibition and the calculated $\Delta G_{binding}$ values for each compound yielded a correlation $R^2$ of 0.68 (Supplementary Fig. 4). As additional controls, we obtained previously developed, commercially available Wnt inhibitors and tested them in the Wnt luciferase assay (Supplementary Fig. 5). Inhibitors upstream of β-catenin (porcupine and tankyrase) had no effect, while those targeting β-catenin/TCF (iCRT3, LF3) or β-catenin/CBP (PRI-724) did inhibit the luciferase reporter although with a weaker potency than compound **13**.

**Macrocycle 13 inhibits Wnt and AR signaling**. We next characterized the dose response curves for compound **13** in the Wnt and AR luciferase reporters. The linear version of **13** (linear 13: NH₂-|N-mpp, N-dmp, N-ipp, N-Smpe, N-Smpe, D-Ala|-OH) was used as a control compound. It incorporates the identical chemical moieties as its cyclic counterpart, but is likely not constrained in an optimal binding conformation and was therefore anticipated to demonstrate weaker affinity. Indeed, as observed in luciferase assays (Fig. 2c), the $IC_{50}$ for Wnt luciferase inhibition was 0.105 μM ± 0.040 (±standard deviation of three replicates) for **13** compared to 3.27 μM ± 0.99 for the linear oligomer. Compound **13** also showed better inhibition of the AR luciferase reporter ($IC_{50}$ of 1.02 μM ± 0.20) compared to **linear 13** ($IC_{50}$ of 7.63 μM ± 1.22). While these values may suggest that the compounds inhibit Wnt signaling with increased potency relative to the AR pathway, the experimental conditions used with each reporter, including siRNA knockdown of APC to activate the Wnt reporter and DHT treatment to activate AR signaling, confound direct comparison of the values. To confirm that **13** does inhibit the β-catenin:TCF protein interaction, we developed a sandwich ELISA that detected recombinant β-catenin binding to a GST fusion protein containing the TCF4 N-terminal β-catenin binding domain (Fig. 3a). In this assay, **13** inhibited binding with an $IC_{50}$ of 5.44 μM±0.82 (Fig. 3b). To test direct binding, we pre-formed dynamic light scattering (DLS) with β-catenin in the presence of vehicle, compound **13**, and α-cyclodextrin. A significant shift in the measured hydrodynamic radius was detected with the addition of compound **13**, indicative of a conformational change associated with a binding event (Fig. 3c). Addition of a similarly sized macrocyclic compound, α-cyclodextrin, was evaluated as a negative control and did not cause the same shift (additional DLS data presented in Supplemental Table 3). Inhibition of the β-catenin:TCF4 protein

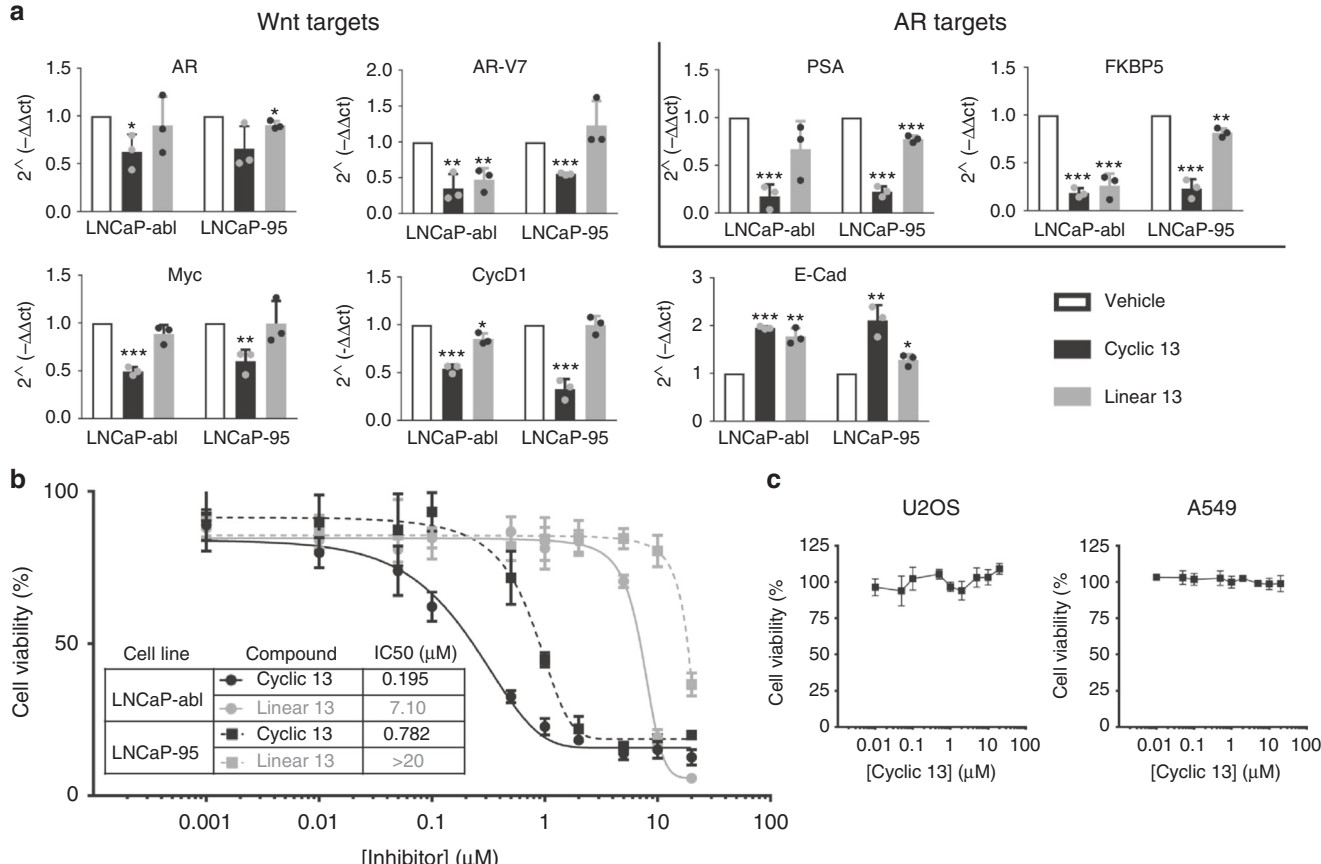

**Fig. 4** Macrocycle **13** affects prostate cancer cell lines. **a** RT-qPCR of mRNA collected from LNCaP-abl or LNCaP-95 cells treated with 10 μM **13** (black) or 10 μM **linear 13** (grey) oligomer for 24 h. **b** Cell viability measurements using CellTiter-Fluor reagent with 5 day oligomer treatment in prostate cancer cell lines. **c** Cell viability measurements with cyclic **13** in two non-prostate, Wnt-independent cell lines. Data are presented as mean and SD ($n = 3$) of three independent experiments (**a**) or representative of three independent experiments (**b**, **c**). Statistical differences for RT-qPCR were calculated by unpaired $t$-tests, $*p < 0.05$, $**p < 0.01$, $***p < 0.001$

interaction was also confirmed in cell culture. HEK293 cells were transiently transfected with a plasmid expressing full length, myc-tagged TCF4. After 48 h treatment with either vehicle (DMSO) or 10 μM **13**, cells were lysed and myc-TCF4 protein was immunoprecipitated. Western blotting was performed, revealing that endogenous β-catenin was co-immunoprecipitated. The β-catenin:TCF association was decreased in cells treated with **13** compared to vehicle treated cells (Fig. 3d).

**Macrocycle 13 inhibits prostate cancer cell growth.** To test the effect of **13** in a model of prostate cancer, we used the LNCaP-abl and LNCaP-95 cell lines. These cell lines were derived from androgen-sensitive LNCaP cells, and passaged in androgen-deprived media to mimic the development of castration resistance in patient tumors[54,55]. We first isolated mRNA from LNCaP-abl and LNCaP-95 cells treated with vehicle, 10 μM **13** and **linear 13** for 24 h to check the expression of endogenous Wnt and AR target genes (Fig. 4a). The mRNA of Wnt target genes cMYC, Cyclin D1, and the AR were all decreased. Transcription of E-cadherin, which has been found to be repressed by Wnt signaling, increased following treatment by **13**. AR target genes PSA and FKBP5 mRNA were also decreased. In addition, the transcription of an AR splice variant, AR-V7, which lacks the ligand binding domain and as a consequence is constitutively active, was also decreased. AR-V7 is particularly challenging to target as most AR antagonists target the ligand binding domain. Similar to experiments discussed above, **13** again showed

increased potency in transcriptional modulation compared to its linear analog. To check if the changes in Wnt and AR target gene expression translated into effects on growth, the viability of LNCaP-abl and LNCaP-95 cells was determined after 5 days of treatment (Fig. 4b). Compound **13** inhibited proliferation with an $IC_{50}$ of 195 nM ± 52 (LNCaP-abl) and 782 nM ± 34 (LNCaP-95). We also tested compound **13** in non-prostate cell lines that have been shown to be Wnt independent (Fig. 4c)[46,47]. Compound **13** had no effect on the proliferation of U2OS (osteosarcoma) and A549 (lung adenocarcinoma) cells.

**Growth of prostate cancer spheroids slowed by macrocycle 13.** To evaluate the impact of the compounds under 3D cell culture conditions thought to be more representative of tumors, we grew LNCaP-abl cells in low attachment plates. The effects on spheroid formation and growth were evaluated following treatment with vehicle, **13**, or **linear 13** administered at the time of seeding at low cell density in a 384-well plate. In a separate experiment, to test the impact on pre-formed spheroids, cells were treated 5 days after initial seeding. Spheroids from both experiments were grown for 22 days, with compound refreshed on days 5, 9, 13, 16, and 20. Figure 5a shows representative images of cells treated with vehicle, 2 μM **13**, or 2 μM **linear 13** at 20 days. PrestoBlue was used to measure cell viability on day 22 (Fig. 5b). Compound **13** reduced spheroid proliferation with an $IC_{50}$ of 0.694 μM ± 0.037 when added in the initial seeding (Day 0). It also reduced spheroid proliferation when

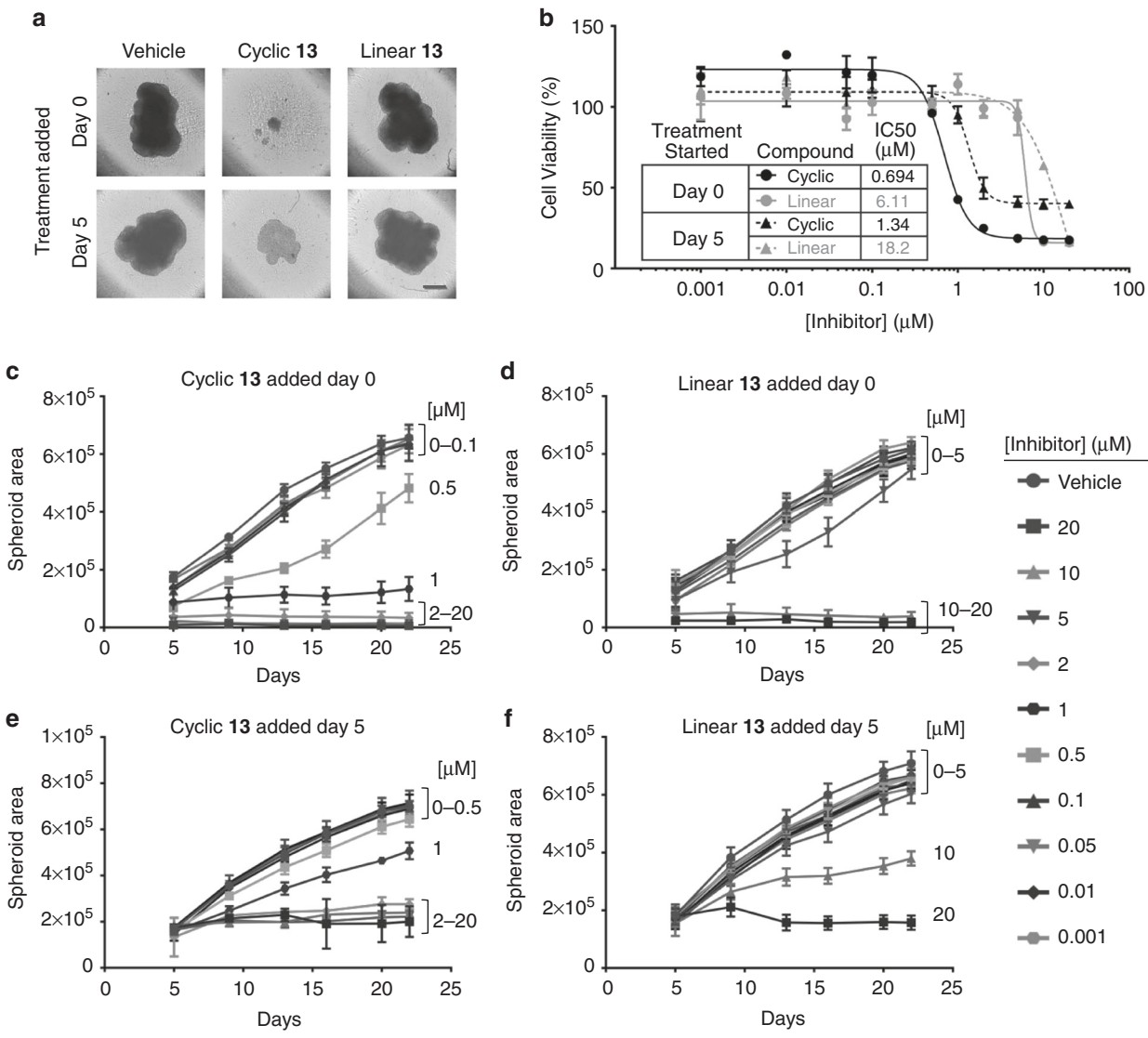

**Fig. 5** Macrocycle **13** slows prostate cancer spheroid cell growth. **a** Representative images of tumor spheroids imaged from above on day 20 and treated with vehicle or 2 μM oligomer. Scale bar represents 400 μm. **b** End point viability measurements taken on day 22 using PrestoBlue. **c–f** Time course of spheroid areas calculated from images taken throughout the experiment. Data are presented as mean and SD ($n = 5$ (**c–f**), $n = 2$ (**b**))

added on day 5, with somewhat lower potency and efficacy and an $IC_{50}$ of 1.34 μM ± 0.11. Figure 5c–f depicts spheroid area over time based on size measurements from the images taken throughout the experiment. Higher concentrations of **13** (20 μM, 10 μM, 5 μM, 2 μM, and 1 μM) prevented spheroid formation when added on day 0 with seeding and 0.5 μM **13** also slowed growth (Fig. 5c). Concentrations of **13** between 0.001 μM and 0.1 μM had little effect. In the case of compound addition to previously formed spheroids (treatment started at day 5), concentrations of **13** at or above 2 μM stalled the growth of preformed spheroids but caused less of a decrease in spheroid area (Fig. 5e) than addition at day 0. **Linear 13** had similar effects with significantly reduced potency (Fig. 5d, f).

**Rescue of eye development in Wnt-activated zebrafish embryos**. To evaluate the efficacy of **13** in vivo, we utilized a *Danio rerio* (zebrafish) model. The relevance of the model for our studies is that zebrafish embryos with Wnt activating mutations in Axin1, a component of the β-catenin destruction complex, fail to develop eyes and a forebrain[56]. The phenotype can be recapitulated by treatment with an inhibitor (termed **BIO**) for another β-catenin destruction complex member, GSK3, in embryos at 6 h post-fertilization (hpf)[57]. It would then be anticipated that **13** could inhibit the chemically overactivated Wnt signal, and potentially rescue eye and forebrain development. To test this, **BIO** was added to zebrafish embryos at 6 hpf with and without **13** as described in Fig. 6a. Indeed, **13** inhibited Wnt signaling as anticipated, as only one of the eight fish treated with GSK3 inhibitor **BIO** alone developed eyes, while all the fish treated with both **BIO** and **13** developed eyes (Fig. 6b–d). This result was reproduced in 2 additional independent experiments each showing that 0/8 **BIO** treated fish developed eyes while 8/8 **BIO** plus **13** treated fish developed eyes. In addition, linear **13** was tested and in each replicate 6/8 fish developed eyes although most of the eyes appeared smaller and less developed (Supplementary Fig. 6a). This result is consistent with cell culture experiments showing cyclic **13** is more potent than linear **13** (Figs. 4, 5). Images of replicate experiments are shown in Supplementary Fig. 6b.

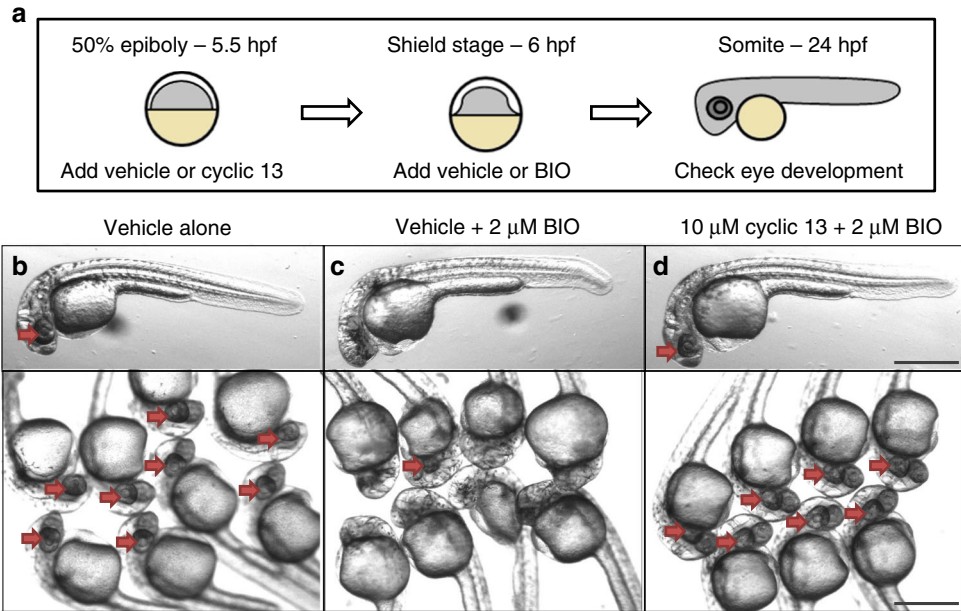

**Fig. 6** Zebrafish phenotypic assay of Wnt signaling. **a** Schematic for experimental design. **b–d** Images of 24 h post-fertilization of zebrafish embryos treated with vehicle alone (**b**), vehicle plus **BIO** to activate Wnt signaling (**c**), or macrocycle **13** in combination with **BIO** (**d**). Red arrows point to developing eye structures. Scale bar represents 500 μm

## Discussion

The Wnt pathway remains a tantalizing but elusive target for drug discovery[38]. Chemical inhibition of Wnt signaling has been achieved through multiple strategies. For example, Wnt secretion and function necessitates the post-translational addition of a palmitoyl group, catalyzed by the acyltransferase porcupine. Enzymatic inhibitors of porcupine have been developed and shown to inhibit the secretion of Wnts[58,59]. In addition, antibodies against frizzled receptors have also been developed to inhibit Wnt signaling at the cell surface[60]. Both of these methods have therapeutic potential for tumors bearing mutations in R-spondin or the ubiquitin ligases ZNRF3 and RNF43, which cause amplification of Wnt signals. Clinical trials for such Wnt inhibitory agents have been initiated. Nevertheless, the majority of activating Wnt mutations occur in the APC gene or the phosphorylation domain of β-catenin, downstream of both porcupine and frizzled. Thus, it is desirable to identify inhibitors downstream of these mutations. Multiple interactions have been described between β-catenin and components of the Wnt enhanceosome on promotors of Wnt target genes[61]. The central interaction between β-catenin and TCF is critical to its Wnt activation and represents an attractive target for the general inhibition of canonical Wnt signaling.

In this study, we demonstrate how computational tools can facilitate the design of oligomers that target β-catenin and disrupt its interaction with TCF. We evaluate peptoid–peptide hybrid macrocycles designed to fold into structures complementary to a region on the β-catenin surface that interacts with TCF. We demonstrate that an oligomeric macrocycle exhibits potent antiproliferative effects on prostate cancer cell lines in both 2D and 3D models. In vivo efficacy of this compound (**13**) is established in a zebrafish model of Wnt signaling. The hypothesis that the conformational constraint imposed on the macrocyclic oligomer enforces steric and chemical complementarity for the β-catenin surface is strongly supported by comparisons to a linear analog, which exhibits markedly diminished activities in vitro and in vivo.

In our proposed mechanism of action, the oligomer macrocycle inhibits canonical Wnt target gene expression via inhibiting the interaction between β-catenin and TCF. We also observe decreased AR pathway target gene expression, which could result either from direct inhibition of the β-catenin:AR interaction or indirectly through the decrease in AR gene transcript levels upon Wnt signal inhibition. It remains to be shown which pathway when inhibited, Wnt or AR, plays a critical role in the antiproliferative phenotype. There may be a synergistic effect of inhibiting both pathways concurrently.

The β-catenin:TCF PPI has been targeted previously[38,46,47]. However, there does not appear to be rapid clinical implementation of this pharmacological approach[38]. An example of possible difficulties is exemplified by iCRT3 whereby the compound was not systemically stable, requiring direct tumor injection in order to attain growth inhibition in mouse xenograft experiments[62]. In contrast, peptoids have been shown to be stable to proteolytic cleavage[8,9]. Additionally, peptoids have been shown to exhibit generally improved membrane permeability compared to the analogous peptide sequences[63,64]. Favorable permeability profiles were shown to be consistent and tolerant to variations in oligomer side chain type[65]. Recently, another hexameric peptoid-peptide macrocycle, designed as a CXCR7 modulator, was demonstrated to exhibit both membrane permeability and oral bioavailability in rats[66]. In addition to pharmacokinetic issues, there is the potential that targeting β-catenin will lead to on-target toxicity[67]. It remains to be seen whether it is possible to achieve a therapeutic window where the inhibitor lowers overactivated Wnt signaling, such as in β-catenin and APC mutated tumors, but does not negatively affect basal Wnt signaling levels in systems such as the intestine and bone that require it for general maintenance. The potential for achieving this balance is supported by our zebrafish experiment, in which compound **13** successfully inhibits the artificially overactivated Wnt signaling pathway but does not otherwise harm embryo development. Additional β-catenin nuclear interactions have also been targeted, such as with BCL9 and CBP[68,69]. The β-catenin/CBP inhibitor PRI-724 has reached the clinic, undergoing trials for colon cancer, AML, and pancreatic adenocarcinoma[70]. Because PRI-724 binds to CBP (and not β-catenin), it may have a different therapeutic profile than a drug targeting

β-catenin:TCF and there may be opportunities for combination therapy with compound **13**.

Future testing with **13** in mouse xenograft models of prostate cancer will be important to determine its therapeutic potential, as demonstrated for other bioactive peptoid oligomers[71]. The ability to introduce a broad range of alternative oligomer side chains should facilitate further improvements of activity along with enhanced pharmacokinetic and pharmacodynamic properties towards the identification of an optimized lead compound. In addition to promising activity versus prostate cancer, macrocycles targeting β-catenin:TCF could also be of therapeutic benefit in other forms of cancer for which Wnt signaling is similarly pro-tumorigenic, such as colon and breast cancers. More generally, this study portends further computer-assisted discovery of increasingly complex folded oligomers to address the vast number of different PPI relevant to human disease.

## Methods

**Synthesis of compounds 1–13.** 2-chlorotrityl resin (Anaspec) was incubated with Fmoc-D-Alanine (Sigma) for 1.5 h followed by washing 5× with DMF. Deprotection was carried out with 20% piperidine in DMF. Peptoid residues were then added with iterative steps of 1.2 M bromoacetic acid + DIC for 20 min and 1 M amine for 1 h. Compound **13** used the following amines: S-(-)-4-methoxy-α-methylbenzylamine (Sigma), 3-isopropoxypropylamine (Sigma), 3,5-dimethoxyaniline (Sigma, overnight incubation), and 2-phenoxyphenethylamine (Sigma). The resin was cleaved with 8:1:1 DCM:Acetic acid:HFIP for 1 h. The DCM was removed via rotary evaporation and the remaining solid was dissolved in 50:50 water:ACN and lyophilized overnight. Reverse phase HPLC was used to purify the linear product. Cyclization was carried out using 25 mg of linear compound with 5X PyBOP and 10X DIEA in dry DCM with less than 1 mg/ml compound overnight under argon. The DCM was removed via rotary evaporation and the remaining solid was dissolved in 50:50 water:ACN and lyophilized overnight. The final product was purified to > 95% by reverse phase HPLC and verified by liquid chromatography-mass spectrometry (Agilent LCMSD Trap XCT). Mass spectrometry peak identified for compound **13** (m/z): $[M + Na]^+$ calcd. for $C_{59}H_{72}N_6O_{12}Na$ 1079.5; found, 1078.346. A key for chemical abbreviations is supplied in Supplementary Table 4.

**Cell culture.** LNCaP-abl and LNCaP-95 cells were cultured in indicator free RPMI 1640 media (Life) supplemented with 10% charcoal stripped FBS, 1% pennicillin/streptomycin, and 1% L-Glutamine. HEK-293 Cells were cultured in DMEM media (Life) supplemented with 10%FBS and 1% pennicillin/streptomycin. A549 and U2OS cells were cultured in RPMI 1640 media (Life) and McCoy's 5 A modified media (Life), respectively, both supplemented with 10%FBS and 1% pennicillin/streptomycin. LNCaP-abl, and LNCaP-95 cell lines were generous gifts from Z. Culig (Innsbruck Medical University, Austria) and J. Isaacs (Johns Hopkins University, Baltimore, MD), respectively. Cell lines were authenticated by STR profiling (Genetica Cell Line Testing; Burlington, NC). A549 and U2OS were purchased from ATCC (product numbers CCL-185 and HTB-96). Cell lines were routinely screened for mycoplasma. Luciferase assays were conducted using stably transfected TOP-flash and 3xARE LNCaP-abl cell lines. For the Wnt pathway, TOP-Flash LNCaP-abl cells were treated with 1 nM siAPC (a 1:1:1 mix of Ambion Silencer Silect siAPC RNA, ID#s s1433, s1434, s1435) for 24 h and seeded into 384-well plates at 5000 cells/well, followed by 24 h of treatment with compound. For the AR-pathway, 3xARE LNCaP-abl cells were seeded into 384-well plates at 5000 cells per well, treated for 48 h with compound, with the addition of R1881 for the final 8 h. Both were measured using the ONE-Glo detection kit (Promega) and data is standardized to CellTiter-Fluor (Promega) viability readings to account for potential variations in well to well cell number. Control Wnt inhibitors were purchased from Sigma with the exception of PRI-724 which was purchased from Selleckchem. Co-immunoprecipitation was carried out using HEK293 cells transfected with either myc-TCF (addgene #32738) or myc-TCF dominant negative (addgene #32739) followed by 48 h of treatment with compound. Cells were lysed using Triton lysis buffer and 1 mg total protein was incubated overnight at 4 C with 40 μl anti-cMyc magnetic beads (Thermo). Total protein concentration in cell lysates were determined using Pierce Rapid Gold BCA Protein Assay Kit (Thermo). The beads were washed 3× with lysis buffer then boiled in 2X sample loading dye. SDS-PAGE and western blotting were performed using anti-Myc (CST 2276) and anti-β-catenin (CST 8480). Both antibodies were diluted in blocking buffer 1:1000 and incubated overnight at 4 C with the membrane. Full blots from IP (Fig. 3d) are depicted in Supplementary Figure 7. RT-qPCR was performed using RNA isolated with the RNEasy kit (Qiagen) from LNCaP-abl and LNCaP-95 cells after 24 h treatment with 10 μM compound. The Verso cDNA synthesis kit (Thermo) with 1 μg total RNA was used to generate cDNA. RNA concentration was determined using a Thermo Scientific NanoDrop 2000 spectrophotometer. RT-qPCR data was collected using Fast SYBR Green Master Mix on a QuantiStudio 6 Flex (Life

Technologies). RT-qPCR data are presented as averages relative to RPL19 Ct values (Δ#1) and DMSO control samples(Δ#2) as relative quantitation ($2^{-\Delta\Delta Ct}$) in triplicate of biological replicates. RT-qPCR primers are given in Supplementary Table 5. 2D proliferation assays were conducted using 384-well plates seeded with 1000 cells/well for LNCaP-abl or LNCaP-95 cells and 1000 cells/well for A549 or U2OS cells. After 5 days of treatment the cell viability was measured with the CellTiter-Fluor Cell Viability Assay (Promega) and data is standardized to vehicle control.

**Recombinant proteins and ELISA.** The ELISA was conducted using recombinant GST-tagged catenin binding domain (CBD) of hTCF4, AA8-54, and His6-β-catenin armadillo domain, AA134-668. Both proteins were expressed in BL21 DL3 (NEB) bacteria grown to $OD_{600}$ 0.6 and induced with 0.5 mM IPTG (Sigma) for β-catenin and 0.1 mM IPTG for TCF. The cells were then shaken for 8 h at 30 C. The bacteria were then pelleted, resuspended in PBS, and lysed with a bacterial FRENCH pressure cell press (Thermo). The lysates were incubated overnight with Ni-NTA or glutathione affixed resin (Thermo), washed with TBS, and eluted using 250 mM imidazole or 20 mM glutathione in 20 mM Tris-HCl pH 8.0 and 200 mM NaCl. The imidazole or glutathione were removed by dialysis into a buffer containing 20 mM Tris-HCL pH 8.0, 200 mM NaCl, and 10% glycerol. Lysates were then aliquoted and frozen at −80 °C. The ELISA was initiated with adsorption of 5 μg/ml GST-TCF CBD into a 96-well Nunc Maxisorp plate (Fisher) overnight at 4 °C. BSA was adsorbed to control wells. The wells were washed, blocked with 5% milk for 1 h, washed, incubated with 5 nM His6-β-catenin with varying concentrations of compound in washing buffer for 1 h, washed, incubated with anti-His6 (Thermo MA1-135) diluted 1:10000 in blocking buffer for 1 h, washed, incubated with anti-mouse-HRP (Santa Cruz) diluted 1:15000 in blocked buffer for 1 h, washed, and then developed using 1-Step Ultra TMB-ELISA (Thermo). His6-β-catenin was pre-incubated with compound for 30 min prior to its addition to the plate.

**Dynamic light scattering.** Dynamic light scattering was measured with a Wyatt DynaPro NanoStar. Aliquots of recombinant frozen His6-β-catenin were thawed fresh for each experiment and centrifuged at high speed for 10 min. Compound 13 or α-cyclodextrin (Sigma) were added to a concentration of 10 μM and compared with the addition of 1% DMSO (vehicle). Ten readings were measured per experiment, each with an integration time of 5 s. Three experiments were performed per condition. Data was collected and analyzed using DYNAMICS software (Wyatt).

**Spheroid 3D cell culture.** LNCaP-abl cells were seeded into 384-well ultra-low attachment plates (Corning), 1000 cells/well. Varying concentrations of compound were added either on the same day as seeding or on day 5. Media was changed every 3–4 days using a Biotek automated media exchanger. Images were taken using an ArraySca VTI (Cellomics) on days 5, 9, 13, 16, 20, and 22. Images were analyzed for spheroid area with the HCS Studio Cellomics Scan Version 6.6.0 (Thermo) software. A final viability measurement was taken on day 22 using PrestoBlue reagent (Thermo).

**Zebrafish experiments.** The zebrafish phenotypic assay was conducted using wild-type zebrafish embryos seeded into a 96-well round bottom plate at 4 hpf in 50 μl of Fish Water (60 mg/l sea salt, Instant Ocean, Blacksburg, VA, USA) supplemented with 1 mg per L Methylene Blue (Sigma-Aldrich). The embryos were not dechorionated. At 50% epiboly, roughly 5.5 hpf, 50 μl of 30 μM **13** or Linear **13** was added. About 30 min later at shield stage, 50 μl of 6 nM (3×) **BIO** was added. The embryos were then visualized for the presence of eyes and imaged at 30 hpf. Zebrafish embryos were randomly assigned to non-blinded treatment groups, none were excluded. Sample size was determined with a power calculation to achieve statistical significance (alpha = 0.05, beta = 0.2) if only 60% of the fish had developed eyes. Zebrafish experiments were approved by the NYULMC Animal Care & Use Program under IACUC Protocol Number: 170105-02. All ethical regulations were complied with.

## Data availability

All relevant data is available from the authors upon request. The X-ray crystallographic coordinates of compound **1** have been deposited at the Cambridge Crystallographic Data Centre (CCDC), under deposition number 1866347. The structure can be obtained free of charge from The Cambridge Crystallographic Data Centre via www.ccdc.cam.ac.uk/data_request/cif

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

## Acknowledgements

We thank Sarah Blosser and Dr. Paramjit Arora for the His6-β-catenin bacterial expression plasmid. We also thank Tenzin Gocha and Dr. Ram Dasgupta for the GST-TCF4 catenin binding domain bacterial expression plasmid. This research was supported by NIH CA112226 (SKL), NIH T32GM007308 (JAS), NIH 5T32CA009161 (JAS), NIH NS069839 (HK), and NSF CHE-1507964 (KK). V.S and E.N are supported by the NIH R01GM126891, Blavatnik Family Foundation, and by the Howard Hughes Medical Institute. The High Throughput Biology Laboratory is supported by NIH/NCI P30CA16087 and NYSTEM Contract C026719. T.W.C. gratefully acknowledges the New York University Biology Department for the Fleur Strand Fellowship as well as the Graduate School of Arts and Sciences for the Horizon Fellowship in the Natural and Physical Sciences. We thank Dr. Chunhua (Tony) Hu at the NYU Department of Chemistry for assistance with X-ray structure determination.

## Author contributions

J.A.S., T.W.C., C.Y., V.S., E.N., H.K., R.B., M.J.G., K.K., and S.K.L. designed research; T.W.C., K.K., and R.B. performed modeling and compound design; T.W.C., J.A.S., and A.K. synthesized compounds; M.H. crystalized compound 1; E.M.B. and J.A.S. performed luciferase screening; V.S. performed the DLS assay; J.A.S expressed and purified proteins, and performed cell culture, ELISA, IP, and zebrafish experiments; J.A.S., T.W.C., M.J.G., K.K., and S.K.L. analyzed data; and J.A.S, T.W.C, K.K., and S.K.L. wrote the paper.

## Additional information

**Competing interests:** The authors declare no competing interests.

