## [Peer Review File · Nature Communications]

Reviewers' comments:

Reviewer #1 (Remarks to the Author):

The manuscript by Kirshenbaum, Logan and coworkers describes the structure-based design of inhibitors of a protein-protein interaction (PPI), namely between beta-catenin and TCF transcription factors. This PPI plays a central role in activated Wnt signaling which has implications in the development of various types of cancer. Using the Rosetta suite, the authors design a cyclic pentameric peptoid structure that inhibits the beta-catenin/TCF interactions and shows activity in cell-based assays as well as zebrafish. The two central research questions addressed here are (i) in silico design of cyclic peptoid ligands and (ii) generation of downstream Wnt signaling inhibitors. Both being of high relevance for the research community.

The manuscript does not cite work highly relevant to this study and it reveals considerable weak points in both of the central questions which would have to be addressed before considering publication in Nature Communications (for details see list below):

(i) To validate the accuracy of the in silico prediction a more thorough investigation of macrocycle binding is necessary (Kd or Ki value, validation of exact binding site and macrocycle orientation).
(ii) Inhibitors of the beta-catenin/TCF interaction have been reported and it should be shown if/what advantages the new one in this manuscript has.

Here is a list of points that have to be addressed before publication:

- 1) The first sentence of the introduction cites a random set of examples (targeting of Ras, MDM2, MCL-1). It would be more informative to cite more general reviews that summarize small molecule and peptidomimetic inhibitors of protein-protein interactions.
- 2) The second part of the first paragraph dealing with the potential advantages of peptide-derived ligands is completely lacking references (see e.g. Matsson and Kihlberg, Med Chem, 2017, 60, 1662)
- 3) Also, when introducing Wnt signaling, an appropriate review should be cited (e.g. Clevers and Nusse, Cell 2012, 149, 1192).
- 4) The introduction and discussion name a few examples of Wnt inhibitors, but the most critical ones that involve peptidomimetic inhibitors of beta-catenin are missing (those involve mice experiments and a crystal structure of one of the inhibitors with beta-catenin!):
 - Takeda et al., Sci Transl Med 2012, 4, 148ra117
 - Grossmann et al., PNAS 2012, 109, 17942
 - Dietrich et al., Cell Chem Biol 2017, 24, 958
- 5) The caption of Figure 1 should state the PDB codes of all structure used to assemble the figure.
- 6) Figure 2b is very speculative and should be removed (may be more suitable as graphical abstract)
- 7) For compound 13 the Kd or Ki value should be determined (e.g. via SPR, ITC or homogenous competition assay such as fluorescence polarization). In addition, the location of beta-catenin binding should be confirmed (TCF has a very extended interaction site) e.g. via competition (or lack of competition) with alternative binding partners of beta-catenin or via mutation of beta-catenin residues. In addition the pendent of an alanine-scan for compound 13 could test if the predicted binding mode is realistic.
- 8) Overall, the methods part is very short and should give more details: e.g. regarding determination of mRNA levels
- 9) Figure captions with respect to cell-based experiments are too brief. It is not clear how many replicates were performed and what errors stand for. Also p-values should be determined to evaluate significance of differences.
- 10) Experiments with zebrafish should contain a control peptide (e.g. linear version) and pictures of the repeated experiments should be shown in SI.
- 11) The discussion should discuss the results in the light of existing inhibitors with the same mode of action.
- 12) The phrase "conformational ordering" is rather unusual it should read: "reducing

conformational flexibility" or "conformational constraint". Also "macrocyclic oligomer" is misleading.

Reviewer #2 (Remarks to the Author):

The paper by Schneider and coauthors reports a set of macrocycles (peptoids) that targets the beta-catenin cleft that is the binding site for the N-terminal TCF beta-hairpin and that inhibits the beta-catenin-TCF interactions, as determined by an in silico guided design. The advantages of peptoids surpass those of conventional small molecules and peptides for targeting protein-protein interactions (PPIs) due to their stable, spatially presented, and sequence-specific structure that mimics protein secondary structure motifs and sufficiently inhibit PPIs, and especially PPIs that harbor broad and flat interfaces like the beta-catenin-TCF interactions. Among these macrocycles, compound 13 was the lead that effectively inhibited Wnt and androgen receptor (AR) in luciferase reporter systems. Further evidence from co-immunoprecipitation and ELISA suggested that the macrocycle disrupted the beta-catenin-TCF4 interactions and the downstream target AR, thereby inhibiting prostate cancer cell growth. The macrocycle in zebrafish models also rescued the eye development failure induced by overexpression of beta-catenin due to BIO treatment. The approach and the presentation of the results and data are clear. However, a few issues need to be addressed.

1. Controls are lacking for the luciferase reporter assays. The controls could be important for comparing the inhibitory potency to previously reported inhibitors targeting beta-catenin-TCF4 interactions.
2. As the authors noted in the manuscript, the macrocycle 13 may directly or indirectly inhibit AR. Testing the macrocycle in cancer cells other than prostate cancer cells is necessary to confirm the inhibitory effect of macrocycle 13.
3. In zebrafish models, no evidence, such as co-IP after treatment, was provided to prove that macrocycle13 in fact acted on the interaction between β -catenin-TCF4.
4. Because the macromolecule features of this class of macrocycles, it would be better to provide some discussion or evidence that they are sufficiently capable of entering cells or nuclei.
5. There are a few typos in the manuscript. On Page 5, line16, "compliment" should be complement; Page 10, line 2, "reported" should be deleted.

Reviewer #3 (Remarks to the Author):

The manuscript "Design of Peptoid-peptide Macrocycles to Inhibit the β -catenin:TCF Interaction in Prostate Cancer" by Schneider et al is an interesting example of rational design applied to peptidomimetic protein-protein interaction inhibitors. The interdisciplinary team consists of experts in peptidomimetic design and androgen receptor biology. They adequately show antagonism of the Wnt pathway in cell culture and in zebrafish, consistent with inhibition of beta-catenin. The ultimate result, a macrocyclic peptide/peptoid hybrid with sub-micromolar activity in disruption of Wnt signaling, is exciting and original. The paper is relatively clear and well-written, the experimental design is sound and the data presented is well-controlled and properly interpreted. The results will be interesting to the wider community of drug designers, as well as to those working more specifically on peptides and peptidomimetics. This manuscript should be published in Nature Communications, but would be strengthened if the following concerns were addressed:

Clearer demonstration of direct inhibition of beta-catenin, and structure-activity relationships: The manuscript presents data from ELISAs and pull-downs that show the macrocycle inhibits the interaction between TCF proteins and beta-catenin. However, this is a well-defined protein-protein interaction that has proven amenable in the past to several other, more direct assays, including

fluorescence polarization assays using dye-labeled peptides or ITC measurement of direct binding to beta-catenin. It would greatly strengthen the present work to observe direct binding to beta-catenin using one of these assays. This would provide better evidence of a physical binding interaction between the designed macrocycle and beta-catenin, and also provide a dissociation constant instead of an IC50. Implementing such an assay would also allow the authors to more easily get structure-activity data for the compound, which would strengthen evidence for the proposed binding mode.

Demonstration that effects on proliferation are Wnt-dependent: The experiments in cell culture are largely controlled by using the linear form of molecule 13 to rule out nonspecific effects. However, the authors do not provide evidence that the phenotypic effects they observe are Wnt-dependent. To do this, they could compare the macrocycle's effects on cell lines with increased or decreased basal Wnt activation levels. Alternatively, they could compare the effects of knockdowns of genes related to Wnt signaling on the molecule's function, or show cooperative effects with other pharmacological modulators of Wnt/beta-catenin signaling. The point would be show that the effects of the compound are due to beta-catenin inhibition by showing synergy with upstream modulators and lack of effect when downstream modulators are introduced.

Response to Reviewers, Nature Communications

We would like to thank the reviewers for the constructive criticism of our manuscript. We believe that the suggested experiments helped strengthen the value of our work and our conclusions. In this revised manuscript, we added completely new figures 3c and 4c as well as supplementary figures 5, 6, and supplementary table 3.

Reviewers' comments:

Reviewer #1 (Remarks to the Author):

The manuscript by Kirshenbaum, Logan and coworkers describes the structure-based design of inhibitors of a protein-protein interaction (PPI), namely between beta-catenin and TCF transcription factors. This PPI plays a central role in activated Wnt signaling which has implications in the development of various types of cancer. Using the Rosetta suite, the authors design a cyclic pentameric peptoid structure that inhibits the beta-catenin/TCF interactions and shows activity in cell-based assays as well as zebrafish. The two central research questions addressed here are (i) in silico design of cyclic peptoid ligands and (ii) generation of downstream Wnt signaling inhibitors. Both being of high relevance for the research community.

The manuscript does not cite work highly relevant to this study and it reveals considerable weak points in both of the central questions which would have to be address before considering publication in Nature Communications (for details see list below):

- (i) To validate the accuracy of the in silico prediction a more thorough investigation of macrocycle binding is necessary (Kd or Ki value, validation of exact binding site and macrocycle orientation).
- (ii) Inhibitors of the beta-catenin/TCF interaction have been reported and it should be shown if/what advantages the new one in this manuscript has.

>>> Point (i) is addressed under comment 7

>>> Point (ii) is addressed under comment 11.

Here is a list of points that have to be addressed before publication:

1) The first sentence of the introduction cites a random set of examples (targeting of Ras, MDM2, MCL1). It would be more informative to cite more general reviews that summarize small molecule and peptidomimetic inhibitors of protein-protein interactions.

>>> We agree and have modified the citations to include more general reviews (citations 1-3).

2) The second part of the first paragraph dealing with the potential advantages of peptide-derived ligands is completely lacking references (see e.g. Matsson and Kihlberg, Med Chem, 2017, 60, 1662)

>>> We agree and have added references for this section (citations 4-6).

3) Also, when introducing Wnt signaling, an appropriate review should be cited (e.g. Clevers and Nusse, Cell 2012, 149, 1192).

>>> We agree and have cited the suggested general Wnt review, along with an updated 2017 review by the same authors, in the introduction (citations 35 and 36).

4) The introduction and discussion name a few examples of Wnt inhibitors, but the most critical ones that involve peptidomimetic inhibitors of beta-catenin are missing (those involve mice experiments and a crystal structure of one of the inhibitors with beta-catenin!):

- Takeda et al., Sci Transl Med 2012, 4, 148ra117
- Grossmann et al., PNAS 2012, 109, 17942
- Dietrich et al., Cell Chem Biol 2017, 24, 958

>>> These references have been added alongside the previously cited papers that report the development of β -catenin/TCF papers. (Discussion Paragraph 4)

5) The caption of Figure 1 should state the PDB codes of all structure used to assemble the figure.

>>> We have added the PDB codes in the legend of figure 1.

6) Figure 2b is very speculative and should be removed (may be more suitable as graphical abstract)

>>> We removed figure 2b.

7) For compound 13 the K_d or K_i value should be determined (e.g. via SPR, ITC or homogenous competition assay such as fluorescence polarization). In addition, the location of beta-catenin binding should be confirmed (TCF has a very extended interaction site) e.g. via competition (or lack of competition) with alternative binding partners of beta-catenin or via mutation of beta-catenin residues. In addition the pendent of an alanine-scan for compound 13 could test if the predicted binding mode is realistic.

>>> We agree with the reviewers' suggestions that it would be valuable to determine a K_d value for the interaction between compound 13 and β -catenin. Unfortunately, the poor stability of β -catenin has precluded our ability to obtain a reliable data set that would establish a K_d. β -catenin has a strong tendency to precipitate and aggregate. Among our efforts to measure a K_d value, we have conducted Isothermal Titration Calorimetry (ITC) and Microscale Thermophoresis (MST) measurements. Both of these measurements provided data that were consistent with the binding of compound 13 to β -catenin. However, low signal-to-noise ratios and anomalous data indicative of protein aggregation, particularly at

higher concentrations of β -catenin, gave rise to serious concerns regarding reproducibility. Using dynamic light scattering (DLS), we were able to show a change in the hydrodynamic radius of β -catenin with compound 13 that is indicative of a conformational change and a binding event (new Fig. 3c). In addition the DLS data confirmed the creation of large aggregates with the combination of β -catenin and compound 13 at high concentrations (new supplementary Table 3, β -catenin + compound **13** peaks 2 and 3 showing an increase in %mass) that are likely responsible for the signal issues with ITC and MST. Thus while we do not feel it appropriate to report a binding constant for the data, we are confident that the new DLS data in combination with the other experiments support our conclusions regarding the ability of compound 13 to abrogate the β -catenin/TCF interaction. Without a consistent means to measure K_d we feel the best method for confirming the binding mode of compound **13** to β -catenin is co-crystallization, which we are attempting but is a longer term project.

8) Overall, the methods part is very short and should give more details: e.g. regarding determination of mRNA levels

>>> We have expanded the methods section to include more specific experimental details, as noted with highlights.

9) Figure captions with respect to cell-based experiments are too brief. It is not clear how many replicates were performed and what errors stand for. Also p-values should be determined to evaluate significance of differences.

>>> The figure captions have been expanded to detail the replicate information. P-values were determined for the RT-qPCR data (Fig. 4a).

10) Experiments with zebrafish should contain a control peptide (e.g. linear version) and pictures of the repeated experiments should be shown in SI.

>>> Pictures have been added in new supplementary figure 6 of the repeated experiments along with the data concerning the linear version.

11) The discussion should discuss the results in the light of existing inhibitors with the same mode of action.

>>> A more thorough discussion of other Wnt inhibitors, including those targeting β -catenin/TCF interaction, has been added (discussion paragraph 4). In addition we tested multiple previously developed commercially available Wnt inhibitors in comparison to our compound, adding an additional figure (supplementary figure 5).

12) The phrase “conformational ordering” is rather unusual it should read: “reducing conformational flexibility” or “conformational constraint”. Also “macrocyclic oligomer” is misleading.

>>> These terms have been modified.

Reviewer #2 (Remarks to the Author):

The paper by Schneider and coauthors reports a set of macrocycles (peptoids) that targets the beta-catenin cleft that is the binding site for the N-terminal TCF beta-hairpin and that inhibits the beta-catenin-TCF interactions, as determined by an in silico guided design. The advantages of peptoids surpass those of conventional small molecules and peptides for targeting protein-protein-interactions (PPIs) due to their stable, spatially presented, and sequence-specific structure that mimics protein secondary structure motifs and sufficiently inhibit PPIs, and especially PPIs that harbor broad and flat interfaces like the beta-catenin-TCF interactions. Among these macrocycles, compound 13 was the lead that effectively inhibited Wnt and androgen receptor (AR) in luciferase reporter systems. Further evidence from co-immunoprecipitation and ELISA suggested that the macrocycle disrupted the beta-catenin-TCF4 interactions and the downstream target AR, thereby inhibiting prostate cancer cell growth. The macrocycle in zebrafish models also rescued the eye development failure induced by overexpression of beta-catenin due to BIO treatment. The approach and the presentation of the results and data are clear. However, a few issues need to be addressed.

1. Controls are lacking for the luciferase reporter assays. The controls could be important for comparing the inhibitory potency to previously reported inhibitors targeting beta-catenin-TCF4 interactions.

>>> We tested commercially available control inhibitors in the luciferase assay (supplemental figure 5).

2. As the authors noted in the manuscript, the macrocycle 13 may directly or indirectly inhibit AR. Testing the macrocycle in cancer cells other than prostate cancer cells is necessary to confirm the inhibitory effect of macrocycle 13.

>>> We tested the compound in non-prostate cancer cells previously described as not dependent on Wnt signaling. We found the compound had no effect on the cell lines proliferation (U2OS and A549), strengthening the evidence for the compound’s mechanism of action as a Wnt inhibitor. This data is shown in the new figure 4C.

3. In zebrafish models, no evidence, such as co-IP after treatment, was provided to prove that macrocycle13 in fact acted on the interaction between β -catenin-TCF4.

>>> To answer the question of whether the compound inhibits the beta-catenin/TCF4 interaction we conducted both a sandwich ELISA and a co-immunoprecipitation in cells. The phenotypic response of the zebrafish suggests that compound 13 inhibits Wnt signaling. We feel that it is unlikely the Wnt inhibition occurs by a different mechanism *in vivo* and it is beyond our technical abilities to perform a co-IP on zebrafish embryos.

4. Because the macromolecule features of this class of macrocycles, it would be better to provide some discussion or evidence that they are sufficiently capable of entering cells or nuclei.

>>> We have added a discussion of the ability for peptoid compounds to get into cells. (Discussion paragraph 4)

5. There are a few typos in the manuscript. On Page 5, line16, “compliment” should be complement; Page 10, line 2, “reported” should be deleted.

>>> The typos have been corrected.

Reviewer #3 (Remarks to the Author):

The manuscript “Design of Peptoid-peptide Macrocycles to Inhibit the β -catenin:TCF Interaction in Prostate Cancer” by Schneider et al is an interesting example of rational design applied to peptidomimetic protein-protein interaction inhibitors. The interdisciplinary team consists of experts in peptidomimetic design and androgen receptor biology. They adequately show antagonism of the Wnt pathway in cell culture and in zebrafish, consistent with inhibition of beta-catenin. The ultimate result, a macrocyclic peptide/peptoid hybrid with sub-micromolar activity in disruption of Wnt signaling, is exciting and original. The paper is relatively clear and well-written, the experimental design is sound and the data presented is well-controlled and properly interpreted. The results will be interesting to the wider community of drug designers, as well as to those working more specifically on peptides and peptidomimetics. This manuscript should be published in Nature Communications, but would be strengthened if the following concerns were addressed:

1. Clearer demonstration of direct inhibition of beta-catenin, and structure-activity relationships: The manuscript presents data from ELISAs and pull-downs that show the macrocycle inhibits the interaction between TCF proteins and beta-catenin. However, this is a well-defined protein-protein interaction that has proven amenable in the past to several other, more direct assays, including fluorescence polarization assays using dye-labeled peptides or ITC measurement of direct binding to beta-catenin. It would greatly strengthen the present work to observe direct binding to beta-catenin using one of these

assays. This would provide better evidence of a physical binding interaction between the designed macrocycle and beta-catenin, and also provide a dissociation constant instead of an IC50. Implementing such an assay would also allow the authors to more easily get structure-activity data for the compound, which would strengthen evidence for the proposed binding mode.

>>> Please refer to reviewer 1, comment 7.

2. Demonstration that effects on proliferation are Wnt-dependent: The experiments in cell culture are largely controlled by using the linear form of molecule 13 to rule out nonspecific effects. However, the authors do not provide evidence that the phenotypic effects they observe are Wnt-dependent. To do this, they could compare the macrocycle's effects on cell lines with increased or decreased basal Wnt activation levels. Alternatively, they could compare the effects of knockdowns of genes related to Wnt signaling on the molecule's function, or show cooperative effects with other pharmacological modulators of Wnt/beta-catenin signaling. The point would be show that the effects of the compound are due to beta-catenin inhibition by showing synergy with upstream modulators and lack of effect when downstream modulators are introduced.

>>> As suggested we compared the effect of the macrocycle in cell lines with decreased basal Wnt activity and observed an absence of effect of the compound on the cell lines' proliferation (new Figure 4c).

Reviewers' comments:

Reviewer #1 (Remarks to the Author):

The comments of all reviewers have been addressed appropriately. However given the focus of this manuscript, the introduction should inform about relevant previously reported Wnt inhibitors. At least, examples that target the same protein-protein interaction (beta-catenin/TCF) should be mentioned (this includes references 45, 51 and 52).

Reviewer #2 (Remarks to the Author):

The authors have satisfactorily responded to all my questions and made the necessary changes to the manuscript.

Reviewer #3 (Remarks to the Author):

The manuscript has been improved by the authors' changes. While better controls for the cell and zebrafish experiments could still be provided, there are enough controls to conclude that the molecules are indeed causing the observed phenotypic changes. The results showing no effect on non-Wnt-dependent cells support that the compound's effects are due to beta-catenin binding, though these data are not conclusive.

The major piece that is still somewhat lacking is the absence of clear evidence for a discrete, 1:1 interaction between compound 13 and beta-catenin. The DLS data indicate some sort of aggregation event, but are inconclusive with respect to detecting a complex between 13 and beta-catenin. It is understandable that beta-catenin may aggregate in solution, but another assay (such as fluorescence polarization or SPR) should be applied to obtain direct binding data. These types of assays are well-precedented for inhibitors of beta-catenin (see papers by the Grossmann group on stapled peptide beta-catenin inhibitors, for example). The ELISA and pull-down data, while supportive, do not demonstrate a discrete binding event since the compound could act via aggregation or another nonspecific means. Overall, this manuscript would be well-served to provide evidence of discrete binding.

Minor issue: the main text still refers to the model of 13 bound to beta-catenin, Figure 2b, which was removed.

Response to Reviewers, Nature Communications, second revision

Reviewer #1 (Remarks to the Author):

The comments of all reviewers have been addressed appropriately. However given the focus of this manuscript, the introduction should inform about relevant previously reported Wnt inhibitors. At least, examples that target the same protein-protein interaction (beta-catenin/TCF) should be mentioned (this includes references 45, 51 and 52).

>>> We have added a paragraph detailing previously reported beta-catenin/TCF inhibitors (introduction paragraph 7).

Reviewer #3 (Remarks to the Author):

The manuscript has been improved by the authors' changes. While better controls for the cell and zebrafish experiments could still be provided, there are enough controls to conclude that the molecules are indeed causing the observed phenotypic changes. The results showing no effect on non-Wnt-dependent cells support that the compound's effects are due to beta-catenin binding, though these data are not conclusive.

The major piece that is still somewhat lacking is the absence of clear evidence for a discrete, 1:1 interaction between compound **13** and beta-catenin. The DLS data indicate some sort of aggregation event, but are inconclusive with respect to detecting a complex between **13** and beta-catenin. It is understandable that beta-catenin may aggregate in solution, but another assay (such as fluorescence polarization or SPR) should be applied to obtain direct binding data. These types of assays are well-precedented for inhibitors of beta-catenin (see papers by the Grossmann group on stapled peptide beta-catenin inhibitors, for example). The ELISA and pull-down data, while supportive, do not demonstrate a discrete binding event since the compound could act via aggregation or another nonspecific means. Overall, this manuscript would be well-served to provide evidence of discrete binding.

Minor issue: the main text still refers to the model of **13** bound to beta-catenin, Figure 2b, which was removed.

>>> The reference to the model of compound **13** has been removed. We agree that demonstration of 1:1 direct binding between beta-catenin and compound **13** is desirable, but due to experimental issues we have not been able to generate direct 1:1 binding evidence. Our previous response detailed our attempts to demonstrate 1:1 binding with isothermal titration calorimetry (ITC) and microscale thermophoresis (MST), and our dynamic light scattering data suggests a direct binding event but shows aggregation that most likely caused complications with ITC and MST. The reviewer has suggested experiments performed by the Grossmann group with stapled peptides could provide a template for our studies. The Grossmann group labeled their peptide of interest with FITC and conducted binding experiments with fluorescence polarization. They were working primarily with linear peptides onto

which they could add FITC to one end without interfering with the interaction between the compound and beta-catenin. Compound **13** is cyclical and our modeling suggests we are taking advantage of the entire surface of the macrocycle to interact with beta-catenin; addition of a bulky FITC group would likely interfere with the compound 13:beta-catenin interaction. For similar reasons SPR experiments would be challenging as attaching a handle, like biotin, to conduct binding measurements would also likely interrupt the interaction. Further, it is unlikely that we could detect a binding event in SPR with beta-catenin bound to the surface instead due to the relatively small molecular weight of compound **13**.

We believe the experiments included in the manuscript support our claim that compound **13** disrupts the interaction between beta-catenin and TCF. We don't believe that lack of evidence for 1:1 direct binding disparages our other exciting findings for which we have provided strong evidence including experiments we conducted in response to the reviewers initial helpful comments.